# Whole-Genome Association Mapping and Genomic Prediction for Iron Concentration in Wheat Grains

**DOI:** 10.3390/ijms20010076

**Published:** 2018-12-25

**Authors:** Dalia Z. Alomari, Kai Eggert, Nicolaus von Wirén, Andreas Polley, Jörg Plieske, Martin W. Ganal, Fang Liu, Klaus Pillen, Marion S. Röder

**Affiliations:** 1Leibniz Institute of Plant Genetics and Crop Plant Research (IPK), Corrensstrasse 3, D-06466 Stadt Seeland, OT Gatersleben, Germany; eggert@ipk-gatersleben.de (K.E.); vonwiren@ipk-gatersleben.de (N.v.W.); liuf@ipk-gatersleben.de (F.L.); roder@ipk-gatersleben.de (M.S.R.); 2SGS TraitGenetics GmbH, D-06466 Stadt Seeland, OT Gatersleben, Germany; andreas@traitgenetics.de (A.P.); joerg@traitgenetics.de (J.P.); ganal@traitgenetics.de (M.W.G.); 3Institute of Agricultural and Nutritional Sciences, Martin-Luther-University Halle-Wittenberg, Betty-Heimann-Str. 3, 06120 Halle/Saale, Germany; klaus.pillen@landw.uni-halle.de

**Keywords:** Wheat, mineral, iron, GWAS, SNP, candidate genes

## Abstract

Malnutrition of iron (Fe) affects two billion people worldwide. Therefore, enhancing grain Fe concentration (GFeC) in wheat (*Triticum aestivum* L.) is an important goal for breeding. Here we study the genetic factors underlying GFeC trait by genome-wide association studies (GWAS) and the prediction abilities using genomic prediction (GP) in a panel of 369 European elite wheat varieties which was genotyped with 15,523 mapped single-nucleotide polymorphism markers (SNP) and a subpanel of 183 genotypes with 44,233 SNP markers. The resulting means of GFeC from three field experiments ranged from 24.42 to 52.42 μg·g^−1^ with a broad-sense heritability (*H*^2^) equaling 0.59 over the years. GWAS revealed 41 and 137 significant SNPs in the whole and subpanel, respectively, including significant marker-trait associations (MTAs) for best linear unbiased estimates (BLUEs) of GFeC on chromosomes 2A, 3B and 5A. Putative candidate genes such as NAC transcription factors and transmembrane proteins were present on chromosome 2A (763,689,738–765,710,113 bp). The GP for a GFeC trait ranged from low to moderate values. The current study reported GWAS of GFeC for the first time in hexaploid wheat varieties. These findings confirm the utility of GWAS and GP to explore the genetic architecture of GFeC for breeding programs aiming at the improvement of wheat grain quality.

## 1. Introduction

Wheat is the second most produced and consumed food crop worldwide and its products form a fundamental diet in the daily life for people in the whole world (FAOSTAT 2016; http://faostat.fao.org). Wheat grains contain mainly carbohydrates with a small proportion of proteins and essential micronutrients such as iron (Fe) and zinc (Zn) [1,2,3]. Micronutrient deficiency including Fe and Zn are among the most prevalent deficiencies in the developing countries and high-risk groups are women and children [4]. More than 2 billion people are affected with Fe deficiency which has an adverse effect on health, such as retarding the physical growth and affecting the motoric development, leading to fatigue and low productivity [5,6]. Therefore, in regions where the people depend mostly on cereal-based foods, deficiencies in micronutrients become a challenge. On the other side, improving Fe concentrations in the edible part of crops are linked with positive consequences on both grain yield and nutritional status as well as a positive effect on human health [7].

Understanding the genetic basis of Fe concentration in wheat grains is imperative for enhancing Fe values in newly developed varieties. Therefore, we performed a genome-wide association study (GWAS) approach which is one of the main approaches for dissecting complex traits including nutritional quality traits that are controlled by many genes and influenced by the environment [8,9]. Several genetic regions controlling mineral concentration traits in wheat have been identified by applying traditional quantitative trait loci (QTL) analysis using bi-parental mapping populations. For instance, Peleg et al. [10] detected five QTLs on chromosomes 2A, 3B, 5A, 6B and 7A for Fe concentration in a tetraploid wild emmer × durum wheat recombinant inbred lines (RILs) population. Another study identified five QTLs underlying grain Fe concentration (GFeC) in a *Triticum spelta* × *T. aestivum* RIL population, of which three mapped to chromosome 1A while two QTLs mapped to chromosomes 2A and 3B [11]. To our knowledge two GWAS studies have been reported on synthetic wheat lines. Gorafi et al. [12] studied grain iron content in 47 synthetic hexaploid wheat germplasm lines and Bhatta et al. [13] performed GWAS for various grain minerals including Fe on 123 synthetic hexaploid wheat lines. Association mapping for seed Fe content was also performed for other crops such as pearl millet [14] and seed Fe concentration in chickpea [15]; however, no GWAS study on released wheat varieties is available to our knowledge.

Recently, genomic prediction (GP) or genomic selection (GS) approaches were developed based on genome-wide marker information to predict the breeding value of complex traits for which only genotyping data are provided (test population) [16]. These predicted values are called genome estimate breeding values (GEBVs) and are based on actual phenotypic data related to genotypes in a training population [17]. Several methods were adopted for GP or GS calculation such as Bayesian methods, rrBLUP and Genomic best linear unbiased prediction (GBLUP), while the main affecting factor within these methods is the density of the markers [18]. Application of GP will be helpful particularly for complex traits and for traits that are costly to phenotype; therefore, applying GP could speed up the genetic gains in the development of nutrient-dense wheat varieties. To date, numerous plant breeding studies have been published to investigate complex traits such as nutritional quality traits in wheat [19,20].

The goals of this study were (i) to study the natural phenotypic variation of wheat GFeC in a panel of 369 elite wheat varieties grown for three years in the field, (ii) to investigate the genetic architecture of this trait and to identify QTLs by applying a GWAS approach, (iii) to define the gene content in the respective genomic region of the wheat reference sequence as well as to identify potential candidate genes, and (iv) to examine the prediction ability in the present wheat panel by using different statistical models.

## 2. Results

### 2.1. Phenotypic Analysis and Correlations

The analyses of variance (ANOVA) for Fe concentration in grains showed a significant effect of both genotype and years (*p* < 0.001) (Appendix A). Wide genetic variation of GFeCs was found between the genotypes in both the whole panel and subpanel in each year (Appendix A, Appendix A). The genotypic variation of Fe concentrations in each year appeared to be normally distributed (Appendix A). The average of grain Fe based on BLUE values was about 34 μg·g^−1^ dry weight (DW) in the whole and subpanel of genotypes (Figure 1) with a range of 24.42–52.42 μg·g^−1^ DW in the whole panel and 26.99–48.52 μg·g^−1^ DW in the subpanel of genotypes (Figure 1). This trend of GFeC decrease among years may be attributed to environmental effects including rain fall and temperature (Appendix A). In fact, this conclusion was supported by the resulting heritability for Fe concentration across the years which is equal to *H*^2^ = 0.59.

A significant positive correlation ranging from *r* = 0.26 to 0.39 (*p* < 0.001) was found for grain Fe in all three years (Figure 2). As well, a significant positive correlation (0.11–0.26, *p* < 0.001) was present between Fe and thousand kernel weight (TKW) in all three years (Appendix A) and a strong correlation was found between Fe and Zn with values ranging between 0.51–0.68 (*p* < 0.001) over years (Appendix A). In the whole wheat panel, genotype “SW Tataros” showed the highest Fe concentration equaling 52.67 μg·g^−1^ DW based on the BLUEs (Figure 3).

### 2.2. Genetic Analysis and Genes Underlying GFeC Trait

GWAS analysis for the whole panel identified 41 significant MTAs (−log_10_ (*p*-value) ≥ 3) (Figure 4A) which were distributed over the genome with *R*^2^ values ranging between 2.7% to 5.22%. In total 41 MTAs, of which 17 were located on chromosome 3B between 46.6 to 59.8 cM (Appendix A). Due to no common associations among years, our analyses were based on BLUEs by including the most significant 3 SNPs for further analysis (Appendix A). In the subpanel, the number of significant associations including unmapped markers was higher and mounted to 137 MTAs (Figure 5A) with *R*^2^ values ranging from 5.60% to 13.09% (Appendix A). The highest phenotypic variation was related to unmapped markers (AX-158577508 and AX-158577509) and equaled 10.38% and 13.09%, respectively. Fifteen, four and two significant SNPs which were present on chromosomes 2A (763,689,738–765,710,113 bp), 3B (731,263,238–731,264,585 bp) and 5A (538,758,878–539,958,539 bp) were targeted for further analysis.

The QQ plots for SNP results revealed that the distribution of observed association *p*-values were close to the distribution of expected associations (Figure 4B and Figure 5B); that means the model which we implemented for GWAS was sufficiently stringent to control for false positive associations. In a previous study a total of 8 markers in the whole panel and 31 markers in the subpanel (Appendix A) had been found significant for grain Zn concentration in the same germplasm [9]. Based on BLUEs, significant markers from the whole and subpanel were selected for a query against IWGSC RefSeq annotation v1.0 to get their annotations.

In the subpanel, we detected several potential candidate genes that located on chromosome 2A (763,689,738–765,710,113 bp) (Appendix A). Based on the functional annotation, we found genes which encode either a transcription factor (TF) related to the NAC (NAM (no apical meristem)) domain family or a transmembrane protein (Table 1). These genes are well known to play a role in nutrient remobilization in plants [21,22]. Therefore, we conclude that this genomic region harbors several putative candidate genes, which may have a significant role in grain Fe accumulation.

### 2.3. Genomic Prediction of GFeC Trait

GP was evaluated for GFeC trait with three statistical models including GBLUP, ridge regression best linear unbiased prediction (RR-BLUP) and Bayes-Cπ in the whole panel. Prediction ability values were 0.29 to 0.38, 0.27 to 0.35, and 0.20 to 0.35 based on using these methods: GBLUP, RR-BLUP and Bayes-Cπ respectively (Figure 6). The highest value is equal 0.38 (GBLUP) and 0.35 (RR-BLUP and Bayes-Cπ) based on Fe BLUE values. The prediction values within years were almost the same and equaled around 0.2 (Figure 6). Based on the GP results, more accurate estimates of breeding values through marker-based relationship matrices could be obtained by increasing the number of genotypes in the training data.

## 3. Discussion

### 3.1. The Usefulness of the Natural Phenotypic Variation

In human nutrition, the estimated average requirement (EAR) of Fe is 1460 μg/day/person, while the target level for sufficient Fe concentration in wheat grains was established as 52 μg·g^−1^ [23]. In our mapping panel, we observed high phenotypic variation in grain Fe concentrations ranging between 16.77–62.87 μg·g^−1^ among years and identifying 23 lines equal or above the required target (≥52 μg·g^−1^). A similar range of GFeCs was reported for a wheat RIL population (17.8–69.0 μg·g^−1^) which resulted from crossing *Triticum boeoticum* with *Triticum monococcum* [24]. Morgounov et al. [25] found GFeCs in the range of 34–43 μg·g^−1^ for 41 winter w·μg heat cultivars except for one spring wheat cultivar, that had GFeC of 56 μg·g^−1^. Also, the range of Fe concentrations in Indian and Pakistan hexaploid wheat grains was found to be in the range of (9.2 to 49.7 μg·g^−1^) [26]. Therefore, using lines with elevated GFeC are important to develop new varieties for crop improvement.

The heritability of Fe concentration among the years was moderate equaling *H*^2^ = 0.59, suggesting a quantitative nature of inheritance and a considerable environmental influence on the expression of the trait. Gorafi et al. [12] reported a broad-sense heritability value of Fe grain concentration in synthetic hexaploid wheat germplasm of 0.80. Khokhar et al. [27] reported broad-sense heritability equal to 0.75 for Fe grain concentrations in field-grown Indian wheat.

GFeCs showed a significant positive correlation among years (*r* = 0.25–0.38, *p* < 0.001), indicating a relatively stable measurement of the phenotypic data. The resulting correlation values were moderate; that may be attributed to the influence of genetics and environment on the GFe accumulation which can also explain the moderate heritability value (0.59) of GFe. Tiwari et al. [24] found a constant correlation between different locations for grain Fe concentrations which is compatible with our results.

The positive and highly significant correlation between Fe and Zn in addition to a significant positive correlation between Fe, Zn and TKW found in the current study, was also reported in earlier studies in wheat [10,25]. For instance, our results agree with Pandey et al. [26], who reported a positive correlation between GFeCs and Zn concentrations in 150 bread wheat lines. Additionally, Fe and Zn have the same families of transporter proteins in several steps during the transportation from the soil to the grain, for example nicotianamine (NA) related enzymes are important for both of Fe and Zn radial movement through the root [28,29]. As well, such a high correlation between Fe, Zn and TKW suggest that these traits (Fe and Zn) may have the same genetic basis and could be improved simultaneously with TKW or TKW determinants such as starch and protein. Krishnappa et al. [30] found common genetic regions between Fe, Zn, TKW and protein content. Peleg et al. [10] showed a positive correlation between Fe, Zn, TKW and protein content in wheat. Therefore, it is important to shed light on the genetic makeup of these traits together.

### 3.2. Putative Candidate Genes

Based on GWAS analysis, we found that different loci are controlling Fe accumulation in wheat grains indicating that it is a complex trait with polygenic control. In the whole panel, 137 significant associations were underlying grain Fe and were distributed on chromosomes 1A, 1B, 2A, 2B, 3A, 3B, 4A, 5A, 5B, 5D, 6A, 6D, 7B and 7D of which 3 significant SNPs were located on 3B (46.60–47.42 cM). There were no obvious candidate genes detected within the aforementioned region. In the subpanel, three physical regions contained significant SNPs, on chromosomes 2A (763,689,738–765,710,113 bp), 3B (731,263,238–731,264,585 bp) and 5A (538,758,878–539,958,539 bp), but only the 2A region conferred candidate genes involved in iron uptake or homeostasis.

Chromosome 2A conferred six putative genes related to the NAC (NAM (no apical meristem)) domain family proteins (Table 1), which are well known to be involved in accelerated senescence and an increase of nutrient remobilization from leaves to grains. Several studies reported about NAC gene and increasing Fe and Zn content in the grains of wheat [13,21,31,32]. Uauy et al. [21] described that a NAC TF (*NAM-B1*) accelerated senescence and nutrient remobilization from leaves to grains. The reduction in RNA levels of the multiple *NAM* homologs by RNA interference delayed the senescence process and reduced wheat grain protein, Zn, and Fe content by more than 30%. In the same context, Ricachenevsky et al. [31] showed that *NAM-B1* which is one of the NAC TFs has a major role in regulating key genes responsible for the senescence process which leads to higher Fe and Zn concentrations in wheat grains.

Another four genes encoded transmembrane proteins on chromosome 2A. It has been reported that transmembrane proteins are responsible for nutrient uptake in plants and play an important role in enhancing the micronutrient content of grains [22,33]. Therefore, these genes could be important for GFeC in wheat; however, functional characterization studies are required to validate the function of these genes.

### 3.3. Genome-Wide Prediction Accuracy

GP or GS has been proposed as a method to improve the breeding efficiency of quantitative and complex traits. Therefore, we extended our analyses and included GP as a suggested tool for improving a polygenic trait such as GFeC in wheat. Our predictability results showed low to moderate values according to three different years and BLUE values, which agrees with another report that obtained low to moderate predictability values for the macro- and micro-nutrients including Fe in wheat landraces [19]. In spring wheat, GP showed moderate to high prediction accuracy for grain Fe by imputing different statistical models [20]. Based on our findings, GP may be considered as a promising approach for enhancing GFeC in wheat especially when larger size germplasm panels with additional genotypes are used to have more accurate estimates of breeding values.

## 4. Materials and Methods

### 4.1. Plant Germplasm

A population comprised of 369 elite European wheat varieties including 355 genotypes of winter wheat and 14 genotypes of spring wheat, mainly from Germany and France was used in this study. Field experiments were carried out at IPK, Gatersleben, Germany over three consecutive years (2014/2015, 2015/2016 and 2016/2017) using plot with a size of 2 × 2 m for each genotype with six rows spaced 0.20 m apart and more details were described in a previous study by Alomari et al. [9]. The grains were collected randomly from more than 250 plants of each plot to be used in the study. Standard agronomic wheat management practices were subjected without applying fertilizers to the soil.

### 4.2. Milling Process

Three hundred sixty-nine wheat genotypes harvested from three different field experimental trials were prepared for milling process by collecting 50 kernels for each genotype to measure thousand-grain weights (TGW) using a digital seed analyzer/counter Marvin (GTA Sensorik GmbH, Neubrandenburg, Germany). Wheat grains were milled by using a Retsch mill (MM300, Mettmann, Germany), afterward, the whole panel of the milled wheat grains was dried by incubating overnight at 40 °C in the oven.

### 4.3. Iron Concentration Measurements

Fifty mg of dried and milled wheat grain flour was taken to be digested by (2 mL) nitric acid (HNO_3_ 69%, Bernd Kraft GmbH, Germany). The digestion process was completed using a high-performance microwave reactor (UltraClave IV, MLS, Leutkirch im Allgäu, Baden-Württemberg, Germany). All digested samples were filled up to 15 mL final volume with de-ionized distilled (Milli-Q) water (Milli-Q^®^ Reference System, Merck, Germany). Element standards were prepared from Bernd Kraft multi-element standard solution (Germany). Fe as an external standard and Yttrium (Y) (ICP Standard Certipur^®^ Merck, Germany) were used as internal standards for matrix correction. Fe concentrations were measured by Inductively Coupled Plasma Optical Emission Spectrometry (ICP-OES, iCAP 6000, Thermo Fisher Scientific, Dreieich, Germany) combined with a CETAC ASXPRESS™ PLUS rapid sample introduction system and a CETAC autosampler (CETAC Technologies, Omaha, NE, USA).

### 4.4. Statistical Analysis

We used Sigma Plot package 13 to perform the ANOVA and Pearson’s correlation coefficient (*r*) which were calculated for the grain Fe data resulted from the three years. The broad-sense heritability was calculated using the equation:*H*^2^ = *σ*^2^_*G*_/(*σ*^2^_*G*_ + (*σ*^2^_e_/*nE*),where *σ*^2^_*G*_ is the genotype variance, *σ*^2^_e_ represents the variance of the residual and *nE* is the environments number.

Mixed linear model function and the residual maximum likelihood (REML) algorithm were applied to calculate the Best linear unbiased estimates (BLUEs) of Fe concentration in wheat grains for each genotype across the years [34] by considering the genotype as a fixed effect and the environment as a random effect. All these calculations were accomplished using GenStat v18 software (VSN International, Hemel Hempstead, UK).

### 4.5. Genotyping

The whole wheat germplasm (369 varieties) was genotyped using two marker arrays: a 90K iSELECT Infinium array [35] and a 35K Affymetrix SNP array (Axiom^®^ Wheat Breeder’s Genotyping Array, http://www.cerealsdb.uk.net/) [36] and these two arrays were genotyped by TraitGenetics GmbH, Gatersleben, Germany (www.traitgenetics.com). Moreover, a novel 135K Affymetrix array was used to genotype a subpanel of 183 genotypes from the whole genotypes panel [9,37] and this chip was designed by TraitGenetics GmbH. As a reference map, the ITMI-DH population [38,39] was used to anchor the SNP markers of the 90K and 35K chips. The 135K chip markers were genetically mapped on four different F_2_-populations and then physically anchored on the chromosome-based sequence of hexaploid wheat [40].

### 4.6. GWAS Analysis

To identify the MTA and QTL (i.e., genomic regions) for Fe concentration in wheat grains, association analyses were conducted between SNP markers and Fe data for each genotype in each year and BLUEs value. For SNP markers, quality control was applied by considering a minor allele frequency (MAF) ≤3% (equaling 11 varieties out of 369) with rejecting SNPs having missing values or heterozygosity ≥3%, resulting in 15,523 polymorphic SNP markers from both of the 90K iSELECT array and the 35K Affymetrix array and 28,710 polymorphic SNP markers from the 135K Affymetrix array, which were used for association analysis.

GWAS was carried out for Fe concentration data from both panels (whole and subpanel) over individual year plus BLUE values by applying the implemented mixed linear model (MLM) and principal component analysis (PCA) as a correction factor for population structure. Whole wheat genotypes panel was analyzed by using the combination of two SNP chips (90K and the 35K chips) based on their genetic reference map whereas, the subpanel was analyzed by the combination of 90K, 35K and 135K chips which were anchored based on physical locations. The purpose of combining the SNP chips was to increase the density of the used markers, achieving good mapping resolution and to further enhance the power of GWAS output within the germplasm panel. All the detected marker-trait associations (MTAs) above the threshold of −log_10_ (*p*-value) ≥ 3 where considered as a significant MTA.

GWAS analysis was computed based on a MLM and PCA which was used for population correction and stratification by using Genome Association and Prediction Integrated Tool (GAPIT) in R [41]. The appropriateness of the used model was evaluated through *Q*–*Q* plots that were obtained by plotting “expected−log_10_ (*p*-values)” on the *x*-axis and “observed−log_10_ (*p*-values)” on the *y*-axis. The population structure of the investigated genotypes panel was described in a previous study by Kollers et al. [42].

### 4.7. Blasting and Annotation

The significant SNP markers which defined the significant associations underlying GFeC trait were listed to obtain their annotation based on the newly released reference genome sequence of Chinese Spring by blasting their sequance against IWGSC RefSeq annotation v1.0 to detect potential candidate genes [43,44].

### 4.8. Genomic Prediction

#### 4.8.1. GBLUP

We used GBLUP to impute GP for GFeC trait data by using Tassel version 5.2.10 [45]. In this model, we evaluated the prediction accuracy by using fivefold cross-validation with 20 iterations as implemented in Tassel software.

#### 4.8.2. RR-BLUP and Bayes-Cπ

We evaluate the prediction ability with the two GS models that are ridge regression best linear unbiased prediction (RR-BLUP) and Bayes-Cπ [16,46]. For both models, GSs were implemented in R using a fivefold cross-validation as described in previous literature Jiang et al. [47]. Simply, all the individuals were randomly divided into five subsets, in which four of the five were used as estimation set and the remaining one were used as test set. After all the genotypic values of individuals were obtained, we calculate the prediction ability that is the correlation between observed and predicted values. The whole process was repeated 100 times and then the mean value was used as the final prediction ability.

## 5. Conclusions

This study characterized many lines of a diverse wheat panel for GFeCs to understand the natural diversity that exists for Fe grain trait and to identify potential genes that contribute to this phenotypic variation and to examine the prediction accuracy. Broad-sense heritability calculation revealed moderate variation that could be attributed to both genetic and environmental effects. Overall, the resources generated in this study can be used to identify suitable candidate genes for further validation analysis. Results of applying GP models to GFeC showed that the correlation between observed and predicted values was relatively moderate; therefore, it would be useful to study the effects of GxE interactions that may improve the predictability value.

## Figures and Tables

**Figure 1 ijms-20-00076-f001:**
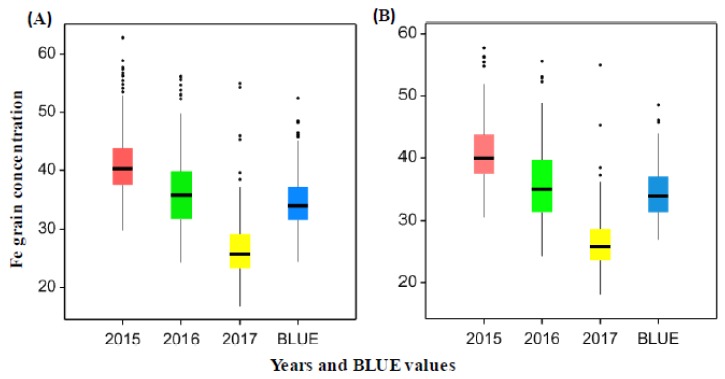
(**A**) Grain iron concentration (µg·g^−1^) in all wheat genotypes of the whole panel for the years 2015, 2016 and 2017 and BLUE values. (**B**) Grain iron concentration (µg·g^−1^) for wheat genotypes in the subpanel for the years 2015, 2016 and 2017 and BLUE values.

**Figure 2 ijms-20-00076-f002:**
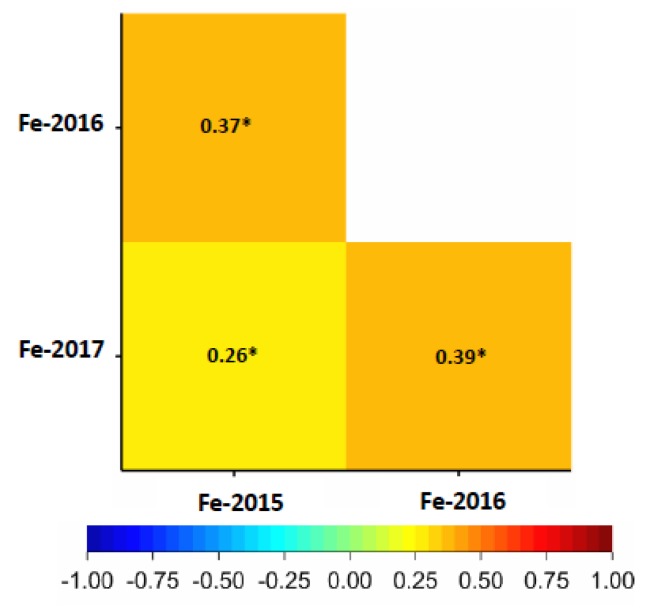
Pearson correlation between Fe grain concentrations (µg·g^−1^) in the years 2015, 2016 and 2017. The degree of significance indicated as * *p* ≤ 0.05.

**Figure 3 ijms-20-00076-f003:**
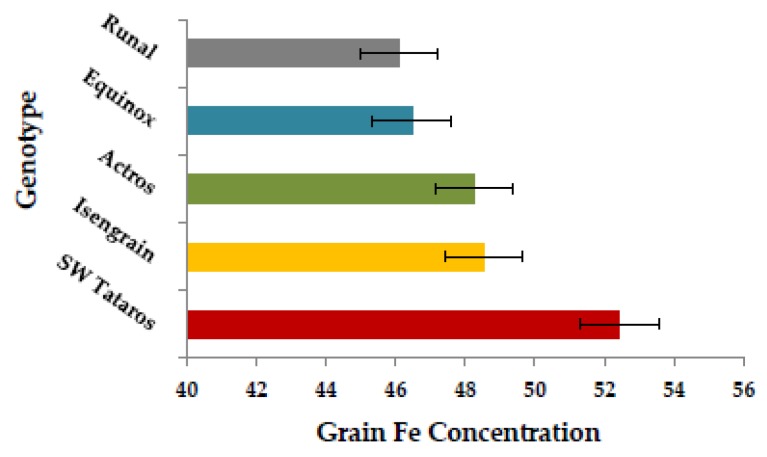
The scale of the top five genotypes with the highest Fe concentration (µg·g^−1^) value crossing years (BLUE).

**Figure 4 ijms-20-00076-f004:**
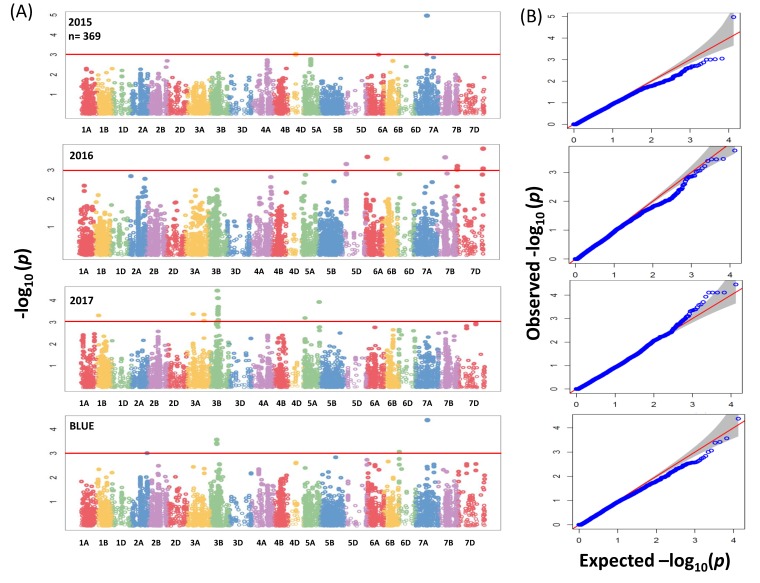
(**A**) Summary of genome-wide association scans for the whole panel of wheat genotypes (369) which were analyzed by using the 90K iSELECT Infinium array and the 35K Affymetrix SNP array for each year (2015/2016/2017) and BLUEs. The horizontal red color line indicated the threshold of −log_10_ (*p*-value) of 3. (**B**) Quantile-quantile scale representing expected versus observed −log_10_ (*p*-value).

**Figure 5 ijms-20-00076-f005:**
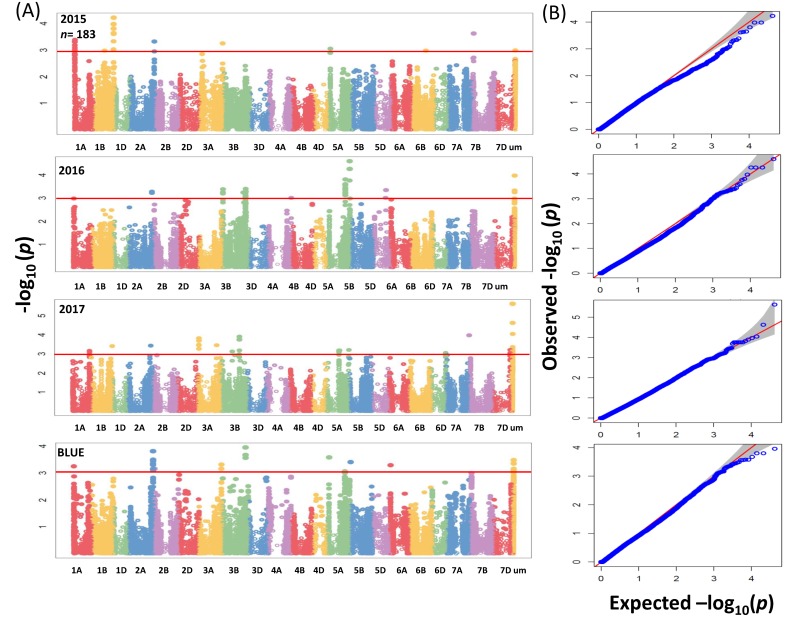
(**A**) Summary of genome-wide association scans for the subpanel of wheat genotypes (183) which analyzed by using the 90K iSELECT Infinium array, the 35K Affymetrix SNP array and the 135K Affymetrix SNP array for each year (2015/2016/2017) and BLUEs. The horizontal red color line indicated the threshold of −log_10_ (*p*-value) of 3. (**B**) Quantile-quantile scale representing expected versus observed −log_10_ (*p*-value).

**Figure 6 ijms-20-00076-f006:**
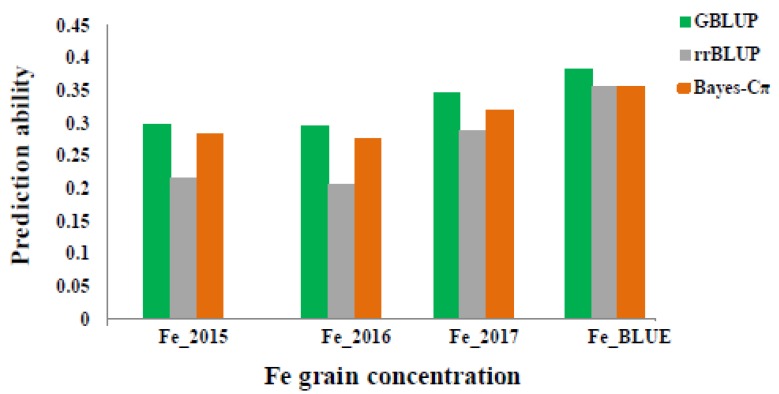
Prediction ability values for grain Fe concentration (µg·g^−1^) according to different years (2015, 2016 and 2017) and BLUEs by using three different statistical models: GBLUP, rrBLUP and BayesC.

**Table 1 ijms-20-00076-t001:** Potential candidate genes underlying GFeC trait in wheat.

Gene ID	Gene Annotation	Chr.	Start (bp)	End (bp)
TraesCS2A01G562600, TraesCS2A01G562700	transmembrane protein, (DUF247)	2A	763,796,420 763,802,755	763,799,183 763,804,683
TraesCS2A01G563600, TraesCS2A01G565000	transmembrane protein, (DUF594)	2A	764,149,111 764,898,033	764,150,898 764,900,078
TraesCS2A01G565900, TraesCS2A01G566000, TraesCS2A01G566100, TraesCS2A01G566200, TraesCS2A01G566300, TraesCS2A01G566400	NAC domain-containing protein	2A	765,277,860 765,373,519 765,392,440 765,441,104 765,514,989 765,546,770	765,278,647 765,375,363 765,393,650 765,442,258 765,518,243 765,547,909

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
