# Peer review of "Whole-Genome Association Mapping and Genomic Prediction for Iron Concentration in Wheat Grains"

_ijms, 2018, doi:10.3390/ijms20010076_

Round 1

Reviewer 1 Report

This manuscript shows very interesting results of the prediction of the concentration of Fe in the wheat grain by the use of GWAS approach.

This is further evidence of the value of a GWAS approach to predict trait of interest and its potential use in breeding programs.

The manuscript presents a lack which concerns the explanation of the correlations between the studied characters. The authors present the correlations without giving a "mechanistic" meaning to this link.

For example Figure 2 shows the correlation between the concentrations of Fe in the grain through the years. These correlation coefficients are small. This character is under the double influence of genetics and environment. This point is not discussed in this manuscript despite its importance. Moreover weather data that could help explain the correlations are not in the manuscript. This is the same pattern that is repeated.

This does not in any way prevent the quality of this article and the results presented which underline the interest of the developed approach.

Author Response

Dear Editor, We are thankful to you and reviewers for having a positive insight into our manuscript ID ijms-406743. In the current revised MS resubmitted we addressed all the arguments and suggestions made by the two reviewers. The changes in the revised manuscript are highlighted yellow. Please see the following sections for a summary of important revisions made to the manuscript based on reviewer’s comments. We hope these changes would be fine with the Editor and Reviewers We look forward to receiving a positive response from you Yours sincerely There is a detailed response to all comments raised by the Reviewers: Comments and Suggestions for Authors This manuscript shows very interesting results of the prediction of the concentration of Fe in the wheat grain by the use of GWAS approach. This is further evidence of the value of a GWAS approach to predict trait of interest and its potential use in breeding programs. The manuscript presents a lack which concerns the explanation of the correlations between the studied characters. The authors present the correlations without giving a "mechanistic" meaning to this link. For example Figure 2 shows the correlation between the concentrations of Fe in the grain through the years. These correlation coefficients are small. This character is under the double influence of genetics and environment. This point is not discussed in this manuscript despite its importance. Moreover weather data that could help explain the correlations are not in the manuscript. This is the same pattern that is repeated. This does not in any way prevent the quality of this article and the results presented which underline the interest of the developed approach. Author’s response: Thank you for your valuable question and suggestion which have been taken in our consideration. Comment 1: For example Figure 2 shows the correlation between the concentrations of Fe in the grain through the years. These correlation coefficients are small. This character is under the double influence of genetics and environment. This point is not discussed in this manuscript despite its importance. Author’s response: In the discussion section/Line 26: Thank you for your suggestion that has been considered in the revised manuscript to explain the resulted correlation values and make the sentence more understandable. Comment 2: Moreover weather data that could help explain the correlations are not in the manuscript. This is the same pattern that is repeated. Author’s response: In the result section line 41-44: we consider your suggestion and added some modifications to the sentence to show that the environment (weather) has an impact on the accumulation of Fe in grains and we added the figure as a supplementary file (Figure S3).

Reviewer 2 Report

The manuscript “Whole genome association mapping and genomic prediction for iron concentration in wheat grains” has been reviewed. The study included two European wheat germplasm pool, and collected three-year phenotypic data (Fe concentration, Zn concentration and TKW) together with genotyping data from three SNP chips. GWAS was performed and candidate genes controlling the traits were proposed without any physical lab-based evidence.

It is an interesting topic worthy an investigation. My only question regarding to experimental design is whether the researchers used same plot for phenotypic data or different which did not show in the manuscript. Other than that, the data collected and presented are supportive to conclusions and overall writing is above average.

Please see attachment for questions and comments.

Round 2

Reviewer 2 Report

The v2 of manuscript "Whole genome association mapping and genomic prediction for iron concentration in wheat grains" has been reviewed. It does have a significant improvement when comparing to v1, and questions/comments are dealt with well.

I would suggest a final proofreading before next submission. 

Thanks